# Striking between-population floral divergences in a habitat specialized plant

**Sumayya Abdul Rahim**[1☯]**, Ullasa Kodandaramaiah**[1☯]***, Aboli Kulkarni**[2¤]**, Deepak Barua**[2]

**1** IISER-TVM Centre for Research and Education in Ecology and Evolution (ICREEE), Indian Institute of Science Education and Research Thiruvananthapuram, Thiruvananthapuram, Kerala, India, **2** Department of Biology, Indian Institute of Science Education and Research, Pune, Maharashtra, India

☯ These authors contributed equally to this work.
¤ Current address: Biodiversity & Paleobiology Group, Agharkar Research Institute, Pune, Maharashtra, India
* ullasa@iisertvm.ac.in

**Data Availability Statement:** All relevant data are within the manuscript and its Supporting Information files.

**Funding:** The work was funded by intra-mural grants from Indian Institute of Science Education

## Abstract

When the habitat occupied by a specialist species is patchily distributed, limited gene flow between the fragmented populations may allow population differentiation and eventual speciation. 'Sky islands'—montane habitats that form terrestrial islands—have been shown to promote diversification in many taxa through this mechanism. We investigate floral variation in *Impatiens lawii*, a plant specialized on laterite rich rocky plateaus that form sky islands in the northern Western Ghats mountains of India. We focus on three plateaus separated from each other by ca. 7 to 17 km, and show that floral traits have diverged strongly between these populations. In contrast, floral traits have not diverged in the congeneric *I. oppositifolia*, which co-occurs with *I. lawii* in the plateaus, but is a habitat generalist that is also found in the intervening valleys. We conducted common garden experiments to test whether the differences in *I. lawii* are due to genetic differentiation or phenotypic plasticity. There were strong differences in floral morphology between experimental plants sourced from the three populations, and the relative divergences between population pairs mirrored that seen in the wild, indicating that the populations are genetically differentiated. Common garden experiments confirmed that there was no differentiation in *I. oppositifolia*. Field floral visitation surveys indicated that the observed differences in floral traits have consequences for *I. lawii* populations, by reducing the number of visitors and changing the relative abundance of different floral visitor groups. Our results highlight the role of habitat specialization in diversification, and corroborates the importance of sky islands as centres of diversification.

## Introduction

Many species are specialized in their ecology—for example they use a small subset of the habitats available to them or interact with only a few species among the many they can potentially interact with. The role of ecological specialization in driving diversification of lineages has been widely appreciated [1–4]. Habitat specialization, wherein a species has evolved adaptations to survive only in a particular habitat (or a few habitats), is one axis of specialization with potentially strong consequences for diversification. When the habitat occupied by a specialized

and Research Thiruvananthapuram and Indian Institute of Science Education and Research - Pune, as well as a grant from the Department of Biotechnology to UK and DB (BT/PR27535/NDB/39/600/2018).

**Competing interests:** The authors have declared that no competing interests exist.

species is patchily distributed, gene flow between the fragmented populations can be restricted, allowing local evolution which can result in divergences between populations, and such divergences can lay the foundation for eventual speciation [5–7].

Species adapted to terrestrial insular habitats—called 'terrestrial islands' or 'inselbergs'—often exhibit high degrees of specialization and habitat fragmentation. 'Sky islands'—rock outcrops and other types of high altitude habitats in mountain systems that are isolated from one another by intervening valleys with markedly different environments [8]—are such terrestrial island habitats. Sky island systems across the world have been shown to be hotspots of diversification [9–21]. Examples of sky island systems include the granitic outcrops of South American Atlantic rainforest [15], the rocky grassland fields (*campos rupestres*) of Brazil, the Madrean woodlands of southern USA and northern Mexico [11], the Eastern Afromontane Biodiversity Hotspot [18] and the tropical montane cloud forests (sholas) in the Western Ghats of India [19].

The Western Ghats mountains also include high altitude laterite rich rocky plateaus—flat topped rock outcrops. These are habitats with extreme environmental conditions such as high temperatures during summer and heavy rains during the monsoon, along with strong wind, high diurnal temperature variation, high evapotranspiration and impermeable soils [22, 23]. The microenvironment in rocky plateau habitats differ from that in the surrounding habitats, making the plateaus terrestrial islands.

Several species of the herbaceous genus *Impatiens* are found on these plateaus. *Impatiens* is a diverse genus, with over 1000 species distributed primarily in the Old World tropics and subtropics [24], with a few species also found in the temperate regions [25]. The genus is the most speciose angiosperm genus in India, with > 220 species, and exhibits exceptional diversity and endemism in the Western Ghats mountains in southern India [26] with more than 120 [27] species, of which > 90% are endemic to these mountains, and new species continue to be reported at a high rate [24, 28–30]. Thus, the Western Ghats are a hotspot of diversification of the genus.

Diversification can begin as divergences in morphological traits, such as floral traits, across populations of a species. The morphological variation between populations of a species can stem from two causes—genetic differentiation and phenotypic plasticity. When the homogenizing effect of gene flow is reduced, adaptations to the local environment, i.e. diversifying selection, can drive genetic differentiation [31]. For animal pollinated plants, differences in pollinators assemblages across populations can result in such diversifying selection [32–34]. Given the importance of floral morphology in mediating attractiveness, and ensuring efficient reward retrieval and accurate pollen transfer [35] floral traits play a critical role in determining pollinator visitation [36, 37], and in turn, pollinators can also influence floral trait evolution.

On the other hand, even in the absence of local adaptation, genetic drift in combination with low gene flow can lead to divergences; the 'isolation-by-distance' model [38]. Thus, reduction in gene flow in conjunction with local adaptation and/or genetic drift may underlie genetic variation between populations. Alternatively, there may be low genetic variation among the three populations, but differences in environments across populations may generate varying phenotypes due to phenotypic plasticity. Many plants display strong variation in floral traits as a result of their sensitivity to edaphic or climatic factors [39], although this plasticity may not be adaptive in all cases.

Here, we investigate diversification of *Impatiens lawii*, a non-cleistogamous hermaphroditic plant specialized on rocky plateau sky islands and endemic to the northern Western Ghats [40, 41]. Our preliminary field observations suggested striking variation in floral morphology across populations (S1 Fig) in the Satara plateau cluster which comprises the Kaas, Thoseghar and Chalkewadi plateaus, all within an altitudinal range of 1145–1230 m above mean sea level. In addition, our extensive field surveys confirmed previous reports that *I. lawii* is absent from

the low-lying areas. The region has been shaped by Deccan trap volcanism ca. 65 million years ago [23, 42]. Although we cannot be certain that the species was historically not more widespread, it is likely that gene flow between the populations has been restricted to some degree, at least in the recent past. Thus, reduction in gene flow in conjunction with local adaptation and/or genetic drift may underlie genetic variation between plateaus in *I. lawii*. Alternatively, the morphological variation observed may be part of the reaction norm to a particular environmental variable (or variables) differing among plateaus. Here we confirm and characterize the variation in floral variation across plateaus, and investigate the factors underlying the patterns of variation.

The objectives of this study were manifold. a) First, using a rigorous quantitative approach, we quantified the extent of between-population floral trait variation in *I. lawii*. In addition to *I. lawii*, we characterized floral variation in *I. oppositifolia*, a congeneric species found on the three plateaus as well as the intervening low-lying regions. Being a relative habitat generalist, *I. oppositifolia* served as a 'control' in our study. These analyses allowed us to test the hypothesis that between population floral differences are stronger in the specialist compared to the generalist species. b) We designed common garden experiments to test the two competing hypotheses—genetic differentiation and phenotypic plasticity. If phenotypic plasticity generates the differences in phenotype across plateaus, there should be no morphological differences under these common garden conditions. On the other hand, if genetic divergences underlie the morphological variation seen, the differences should also manifest under the common garden conditions. c) To test whether floral trait variation across plateaus in *I. lawii* has a functional consequence with respect to pollination, we also quantified floral visitation across plateaus through field surveys. We predicted that floral visitor assemblages differ between populations in the habitat specialist but not in the generalist species, as a consequence a greater variation in floral traits.

## Methods

### Variation in floral traits in natural populations

We sampled 30 individuals each of *I. lawii* and *I. oppositifolia* from every plateau during the peak flowering season (September-October), between 2016 and 2018. Whole individuals were collected and stored in sealed plastic bags and one fully opened flower per individual was preserved in FAA (Formaldehyde Alcohol Acetic Acid, 10%:50%:5% + 35% water). The sampled individuals were selected such that they represented the entire plateau, maintaining adequate inter-individual distance (at least 10m) to minimize the probability of sampling genetically closely related individuals. GPS coordinates were recorded for all individuals sampled.

To characterize variation in floral morphology across populations, we quantified 15 floral traits from each individual, of which 12 were measured (or calculated based on measurements) from dissected floral parts, and three were three-dimensional measurements from fresh flowers at the time of sampling (Table 1; Fig 1A–1C; S2 Fig). Flowers were dissected under a magnifying glass in the laboratory, pasted on a butter paper and scanned immediately using a CanoScan LiDE120 (Canon) scanner at 600 dpi. Floral character measurements were made from these scanned images using ImageJ v 1.47 (NIH Image, Bethesda, MD, USA) [43]. The three floral characters representing the three-dimensional shape of flowers were measured using a digital Vernier caliper with a precision of 0.01 mm.

### Transplant and germination experiments

We conducted two common garden experiments in the greenhouse of Indian Institute of Science Education and Research—Pune, between 2017 and 2018. In the first (hereafter

**Table 1. List of morphological characters studied in populations of *Impatiens species* in the wild and in common garden experiments.** Organ level traits, the three dimensional measurements of the flower made from fresh flowers and sub organ level traits indicate measurements quantified from dissected floral parts. All variables were measured in millimetre (mm) scale.

| | Code | Character | Description | Association |
|---|---|---|---|---|
| a) Organ-level traits | | | | |
| | FL | Flower length | Tip of keel petal to tip of wing petal | Flower size, Pollinator attraction |
| | FW | Flower width | Between two wing petals at its broadest part | Flower size, Pollinator attraction |
| | FOP | Flower Opening | From tip of standard petal to base of wing petal | Flower size, Pollinator efficiency |
| b) Sub-organ-level traits | | | | |
| | WPL | Wing petal length | At its longest part | Flower size, Pollinator attraction |
| | WPW | Wing petal width | At its broadest part | Flower size, Pollinator attraction |
| | STPL | Standard petal length | At its longest part | Pollinator efficiency |
| | STPW | Standard petal width | At its broadest part | Pollinator efficiency |
| | LSPL | Lateral sepal length | At its longest part | Pollinator efficiency |
| | LSPW | Lateral sepal width | At its broadest part | Pollinator efficiency |
| | OVL | Ovary length | At its longest part | Fertilization/Reproduction |
| | OVW | Ovary width | At its broadest part | Fertilization/Reproduction |
| | PDL | Pedicel length | At its longest part | Pollinator attraction |
| | LPL | Lip length | At its longest part | Pollinator efficiency |
| | LPW | Lip Width | At its broadest part | Pollinator efficiency |
| | SPRL | Spur length | At its longest part | Pollinator efficiency |

'Transplant experiment'), we transplanted 30 seedlings per species per plateau in the 4-leaved stage and with a height of 5–8 cm from the three plateaus to 1 litre pots (1 plant per pot) containing 1:1 soil:coco peat in the greenhouse. In the second (hereafter 'Germination experiment'), we used seeds collected from the wild populations in 2016. Seeds were collected from 30 plants each of *I. lawii* and *I. oppositifolia from each plateau*, and stored in paper envelopes at room temperature. Seeds were germinated by placing 3–5 seeds each from an individual on a 1% agar plate and incubated at room temperature. Successfully germinated seedlings were transferred to 1 litre pots (1 plant per pot) and 30 seedlings propagated making sure that every seedling came from a different maternal individual. These seedlings were maintained in a greenhouse that received natural sunlight and had a temperature range of 26 to 30˚C, while relative humidity varied according to local conditions. Characterization of floral trait variation across treatments in both the Transplant and Germination experiments followed protocols used for natural populations (described in the previous section).

## Quantification of floral visitors

Floral visitors to *I. lawii* and *I. oppositifolia* were observed directly in Kaas and Chalkewadi for three days on each plateau during the peak flowering season in September 2017. Observations were performed simultaneously for each species by paired sets of observers. Observations were made during two time periods of every day, between 0800–1300 hours and 1400–1800 hours, and observation sessions were evenly distributed through these time periods. The same set of observers were used and observers alternated between the two *Impatiens* species to minimize bias. Both species form mono-specific patches, and for every observation time point, each set of observers selected a 1 m x 1 m plot for each species. For every observation time point, ten one-minute scans were performed for a total of 10 minutes during which the identity and the number of flower visitors were recorded [44]. All insects that physically touched the flower were counted as floral visitors. Bees were identified to the level of species; butterflies, moths

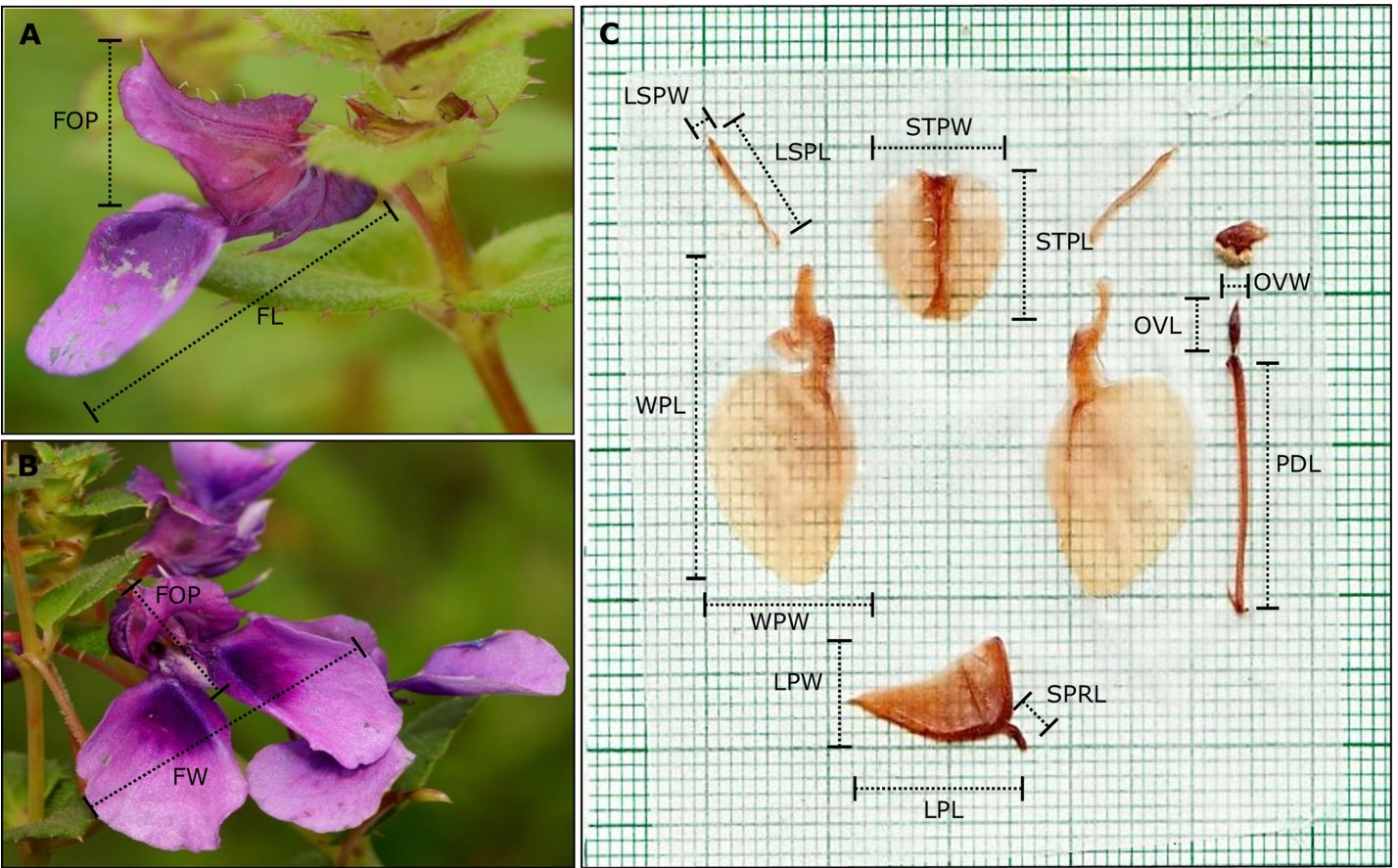

**Fig 1. Morphological characters studied in *Impatiens lawii* flowers.** (A) Side view of the whole flower and (B) Front view of the flower, depicting the three-dimensional characters measured. (C) Quantification of dissected floral parts. Refer to Table 1 for the details of character abbreviations.

and Diptera to the level of morphotypes; and beetles to the level of order. The total number of flowers in the plot was counted at the end of each ten minute session. Floral visitation rates were estimated as the number of visitors per unit time. The cumulative observation time for each species on each plateau was between 11 and 12 hours, and the total observation time was 48.5 hours (2910 minutes).

## Statistical analyses

All statistical analyses were performed in R v3.5.1 (R Core Team, 2018) [45], separately for the wild and common garden experiments. We performed Principal Component Analyses (PCA) using the function *prcomp* on the respective correlation matrices to identify the presence of highly correlated variables and possible outliers [46]. The variables were scaled before the analyses [47]. The FactoMineR v2.4 [48] and factoextra v1.0.3 packages [49] were used to extract and interpret the PCA results. We then fitted Generalized Linear Models (GLMs) using the *glm* function in the package mass 7.3–54 [50], with the first three principal components (PC1, PC2 and PC3), that together explained ca. 70% or more of the variance in the data in all cases, as the response variable and plateau as the categorical independent variable. The GLM analysis was followed by pairwise plateau comparisons using Tukey's HSD post hoc tests with the *glht* link function in the package multcomp v1.4–8 [51].

Our floral visitation rate data were over dispersed, with a high number of unvisited flowers and some flowers with a high visitation rates. Hence, to examine the relationship between visitation rate and flower size, we applied a zero-inflated negative binomial regression (ZINB) analysis which is used for count data that exhibit over dispersion and an excess of zeros [52, 53]. This was done using the package pscl v1.5.5 [54]. The data distribution combines the negative binomial and the logit distributions. The response variable included the visitation rates, number of visits contributed by all floral visitors (total visits per minute per plot), and the predictor variables included plateau, plant species and flower number. Including flower number as one of the predictor variables compensates for variance in visitation rate due to overall differences in the number of flowers in the observed plots. Hence, the significance of predictors of interest (plateau and species) are manifest after controlling for the total number of flowers. Pairwise comparisons for the interaction between visitation rates and presentation order for the ZINB were calculated using the *emmeans* function in the emmeans v1.6.0 package [55] and adopting the Tukey method. The above analyses were also conducted separately for visitation rates of the different floral visitor groups.

## Results

### Correlation between floral traits and variation in flower size

Most floral traits were significantly positively correlated with each other in both *I. lawii* and *I. oppositifolia* and in all three experiments (Wild, Transplant and Germination experiments) (S3 Fig). The overall width of the flower (FW) described the broadest measure of the flower, contributed mainly by the width of the two largest and most conspicuous wing petals which presumably are most important in attracting pollinators [56, 57]. FW consistently showed significant positive correlations with all other floral characters in both species, and we use FW to represent flower size variation across plateaus. For *I. lawii* in the wild, the Chalkewadi population had significantly smaller flowers than those from both Kaas (GLM: Estimate = 6.97, z value = 11.55, p < 0.0001) and Thoseghar (GLM: Estimate = 7.07, z value = 11.71, p < 0.0001) (Fig 2A). However, there was no significant difference in flower size between Kaas and Thoseghar (GLM: Estimate = 0.096, z value = 0.158, p = 0.986). In *I. oppositifolia*, pairwise comparisons indicated no significant variation in flower size between plateaus i.e., Kaas- Thoseghar (GLM: Estimate = 0.991, z value = 1.16, p = 0.476), Thosegar-Chalkewadi (GLM: Estimate = − 0.439, z value = − 0.515, p = 0.864) and Kaas- Chalkewadi (GLM: Estimate = − 1.430, z value = − 1.676, p = 0.214) (Fig 2B). The patterns in the wild were mirrored in both the Germination and Transplant experiments. Thus, for *I. lawii*, flowers from the Chalkewadi population were smaller than those from Kaas and Thoseghar in both common garden experiments, but there were no differences between Kaas and Thoseghar (S1 & S2 Tables). In *I. oppositifolia*, no pairwise comparison was significant for flower size, indicating no variation between populations (S1 & S2 Tables).

**Multivariate analyses.** *i. Floral variation in natural populations.* In the PCA for the 15 floral traits in 90 *I. lawii* individuals, the first principal component axis (PC1) had an eigenvalue of 10.76 accounting for 71.8% of the total variance (S3 Table). This axis was explained mainly by variables related to flower size and pollinator attraction (WPL, WPW, FW, FL), and pollinator efficiency (LPL, LPW). These six variables were highly and negatively correlated with PC1. The second (PC2) and third (PC3) principal components explained 5.1% and 4.7% variation respectively, and were described mainly by pedicel length (PDL) and spur length (SPRL) (Fig 3A; S3 Table). GLMs followed by pairwise comparisons of plateaus based on PC1, PC2 and PC3 axes, which together explained more than 80% of variance, showed that all the three populations differed from each other; Kaas-Thoseghar (GLM: Estimate = − 1.54, z value = − 2.87, p = 0.010), Thoseghar-Chalkewadi (GLM: Estimate = − 6.58, z value = −12.24, p < 0.001),

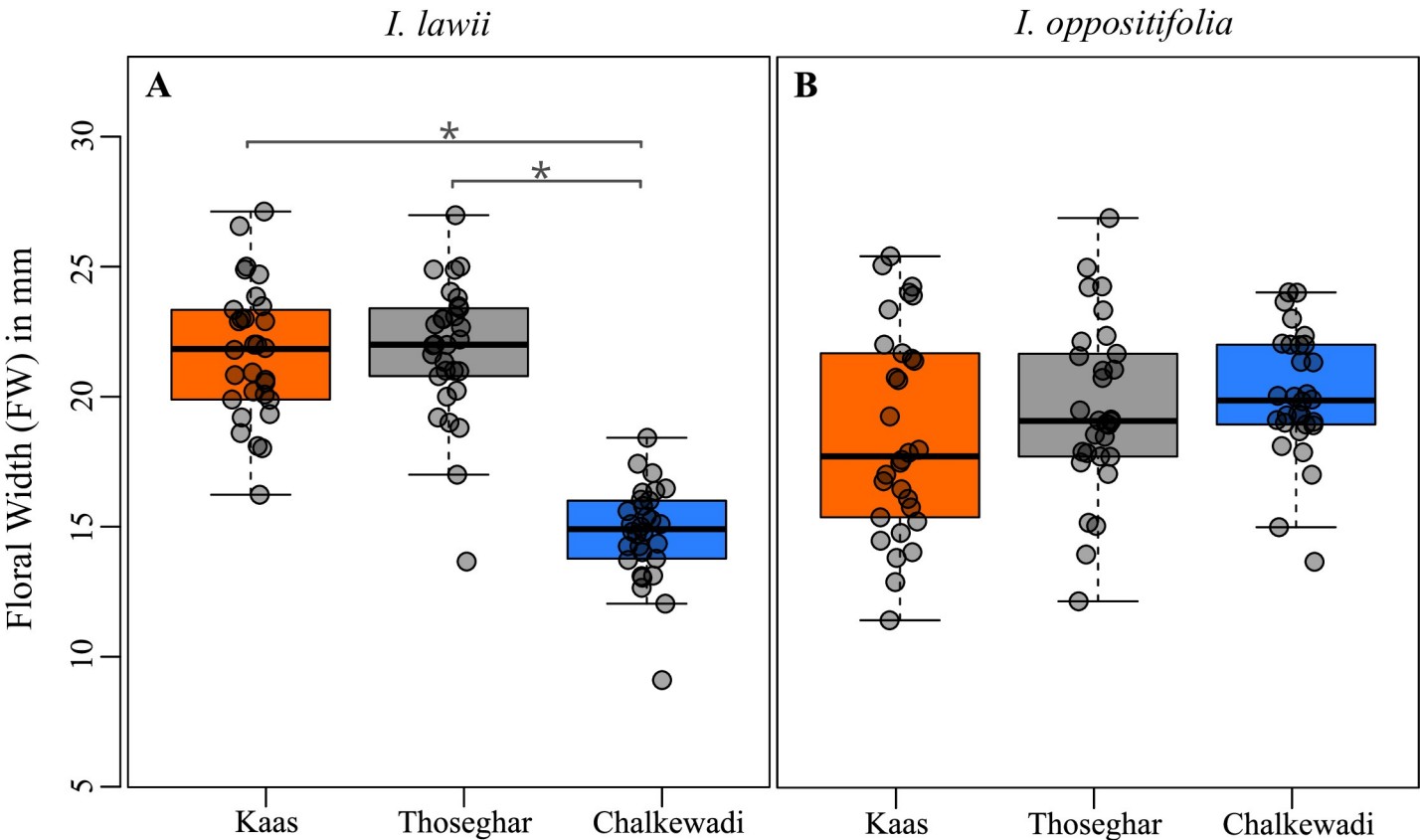

**Fig 2. Flower size represented by flower width (FW) across the three wild populations from the three plateaus.** (A) *I. lawii;* and, (B) *I. oppositifolia*. Boxplots with the median and interquartile range for flower width are presented for 30 individuals from each species and plateau. An asterisk indicates a significant difference (p<0.05) in flower width between plateaus.

Kaas-Chalkewadi (GLM: Estimate = − 5.04, z value = − 9.36, p < 0.001) in *I. lawii* (Fig 4A; S4 Table).

In *I. oppositifolia*, PC1, PC2 and PC3 represented 49.7%, 9.4% and 7.4% of the total variance respectively (S3 Table). The size of the wing petal (WPL, WPW) had the highest contribution to PC1 (Eigenvalue: 7.47) and was negatively correlated with this axis. Ovary width (OVW) and lip width (LPW) contributed mainly to the PC2 variance. Variance in PC3 was described primarily by spur length (SPRL), followed by Pedicel length (PDL) (Fig 3B; S3 Table). Pairwise comparison tests did not discriminate between the three populations; Kaas-Thoseghar (GLM: Estimate = -1.46, z value = -1.80, p = 0.168), Thoseghar- Chalkewadi (GLM: Estimate = − 1.46, z value = − 1.80, p = 0.168), Thoseghar- Chalkewadi (GLM: Estimate = − 0.75, z value = − 0.92, p = 0.622), Kaas-Chalkewadi (GLM: Estimate = 0.70, z value = 0.875, p = 0.656) (Fig 4B; S4 Table).

*ii. Germination experiment*. In the PCA analysis of the 15 floral traits from 90 *I. lawii* individuals in the Germination experiment, the first principal component (PC1) accounted for 70.2% of the total variance (Eigenvalue = 10.6) (S3 Table). As observed in the wild populations, PC1 was explained by variables associated with flower size (WPL, WPW, FL, FW) and pollinator efficiency (LPL, LPW), and were positively correlated with this axis. PC2 and PC3 explained 7.5 and 5.7% of the total variance, respectively. Ovule size (OVL, OVW), pedicel length (PDL) and lateral sepal length (LSPL) mainly contributed to PC2 and PC3 (Fig 3C; S2 Table). GLMs followed by pairwise tests for PC1, PC2 and PC3 together indicated significant

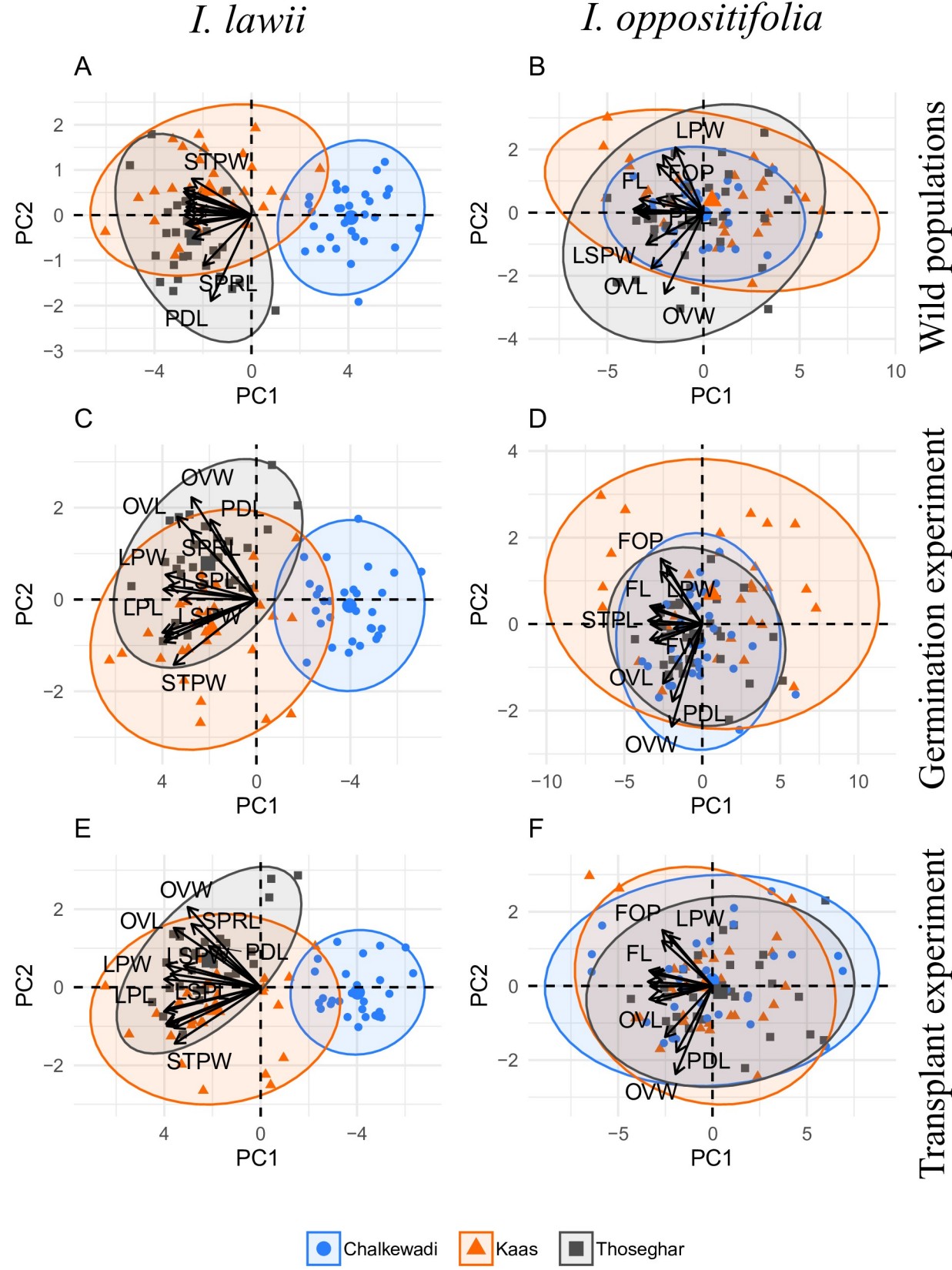

**Fig 3. Principal component analysis of floral morphology.** Biplots are shown for the first two principal component (PC) axes based on the 15 floral traits examined in *Impatiens lawii* (A, C, E) and *Impatiens oppositifolia (*B, D, F), for 30 individuals each from the three plateaus (depicted by the different colours and shapes as shown in the legend). Ellipses indicate 95% confidence intervals. The details of the abbreviations used for floral traits are presented in Table 1.

differences between all the three populations in *I. lawii*; Kaas-Thoseghar (GLM: Estimate = 1.91, z value = 3.99, p = 0.0001), Thoseghar-Chalkewadi (GLM: Estimate = 7.11, z value = 14.81, p < 0.0001), Kaas-Chalkewadi (GLM: Estimate = 5.19, z value = 10.82, p < 0.0001) (Fig 4C; S4 Table).

In the PCA analysis of the 90 *I. oppositifolia* individuals, PC1 (Eigenvalue = 8.76) accounted for 58.39% of the total variance, followed by PC2 (8.26%) and PC3 (6.87%) (S3 Table). PC1 was explained by variables related to flower size (WPL, WPW) and pollinator efficiency (LPL, STPL, STPW), and these variables were all negatively correlated with this axis. Variance in PC2 was mainly contributed by ovary size (OVW) and that of PC3 by spur length (SPRL) and edicel length (PDL). All variables were positively correlated with the respective axis (Fig 3D; S3 Table). The populations did not differ from each other; Kaas-Thoseghar (GLM: Estimate = 0.80, z value = 1.04, p = 0.55), Thoseghar-Chalkewadi (GLM: Estimate = 0.80, z value = 1.04, p = 0.55), Thoseghar-Chalkewadi (GLM: Estimate = 0.44, z value = 0.57, p = 0.83), Kaas-Chalkewadi (GLM: Estimate = – 0.35, z value = – 0.46, p = 0.88) (Fig 4D; S4 Table).

*iii. Transplant experiment.* In the PCA on floral traits of the 90 *I. lawii* individuals in the transplant experiment PC1 (Eigenvalue = 10.63) accounted for 70.8% of the total variance, followed by PC2 (6.5%) and PC3 (5.3%) (S3 Table). PC1 mainly included variables representing flower size (WPL, WPW) and pollinator efficiency (LPL, LPW), and all variables were positively correlated with this axis (Fig 3E; S3 Table). Variance in PC2 was mainly contributed by ovary size (OVW, OVL) and spur length (SPRL), and that in PC3 by pedicel length (PDL). GLMs followed by pairwise tests for PC1, PC2 and PC3 together indicated that the three *I. lawii* populations differed significantly from each other with respect to floral characters; Kaas-Thoseghar (GLM: Estimate = 1.17, z value = 2.769, p = 0.015), Thoseghar-Chalkewadi (GLM: Estimate = 1.17, z value = 2.769, p = 0.015), Thoseghar-Chalkewadi (GLM: Estimate = 7.10, z value = 16.79, p < 0.001), Kaas-Chalkewadi (GLM: Estimate = 5.93, z value = 14.02, p < 0.001) (Fig 4E; S4 Table).

In *I. oppositifolia*, PC1, PC2, and PC3 described 54.3%, 8.1% and 7.1% of the total variance, respectively (S2 Table). Variables related to flower size (WPL) and pollinator efficiency (STPL) contributed to the variance observed in PC1 (Eigenvalue = 8.13). Lip petal width (LPL) and spur length (SPRL) contributed to the maximum variance in PC2, and spur length (SPRL) and ovary size (OVW) to PC3 (Fig 3F; S3 Table). As observed in the wild and the Germination experiment, the three were not differentiated from each other in floral characters; Kaas-Thoseghar (GLM: Estimate = – 0.15, z value = – 0.19, p = 0.980), Thoseghar-Chalkewadi (GLM: Estimate = – 0.80, z value = – 0.92, p = 0.624), Kaas-Chalkewadi (GLM: Estimate = – 0.64, z value = – 0.743, p = 0.738) (Fig 4F; S4 Table).

**Quantification of floral visitors.** The total visitation rates were significantly lower for *I. lawii* in Chalkewadi compared to visitation rates for this species in Kaas (Fig 5A, Table 2). At the same time there was no significant difference in visitation rates to *I. oppositifolia* in the two plateaus.

For Coleoptera and Diptera, visitation rates were lower in Chalkewadi than in Kaas for both *I. lawii* and *I. oppositifolia* (lack of species x plateau interactions, but significant main effect of plateau; Fig 5A, Table 2), implying lower abundance of insects of these two orders in Chalkewadi.

For Hymenoptera (bees), the reduction in visitation rates was particularly pronounced and specific for *I. lawii* in Chalkewadi (significant species x plateau interactions, Table 2,

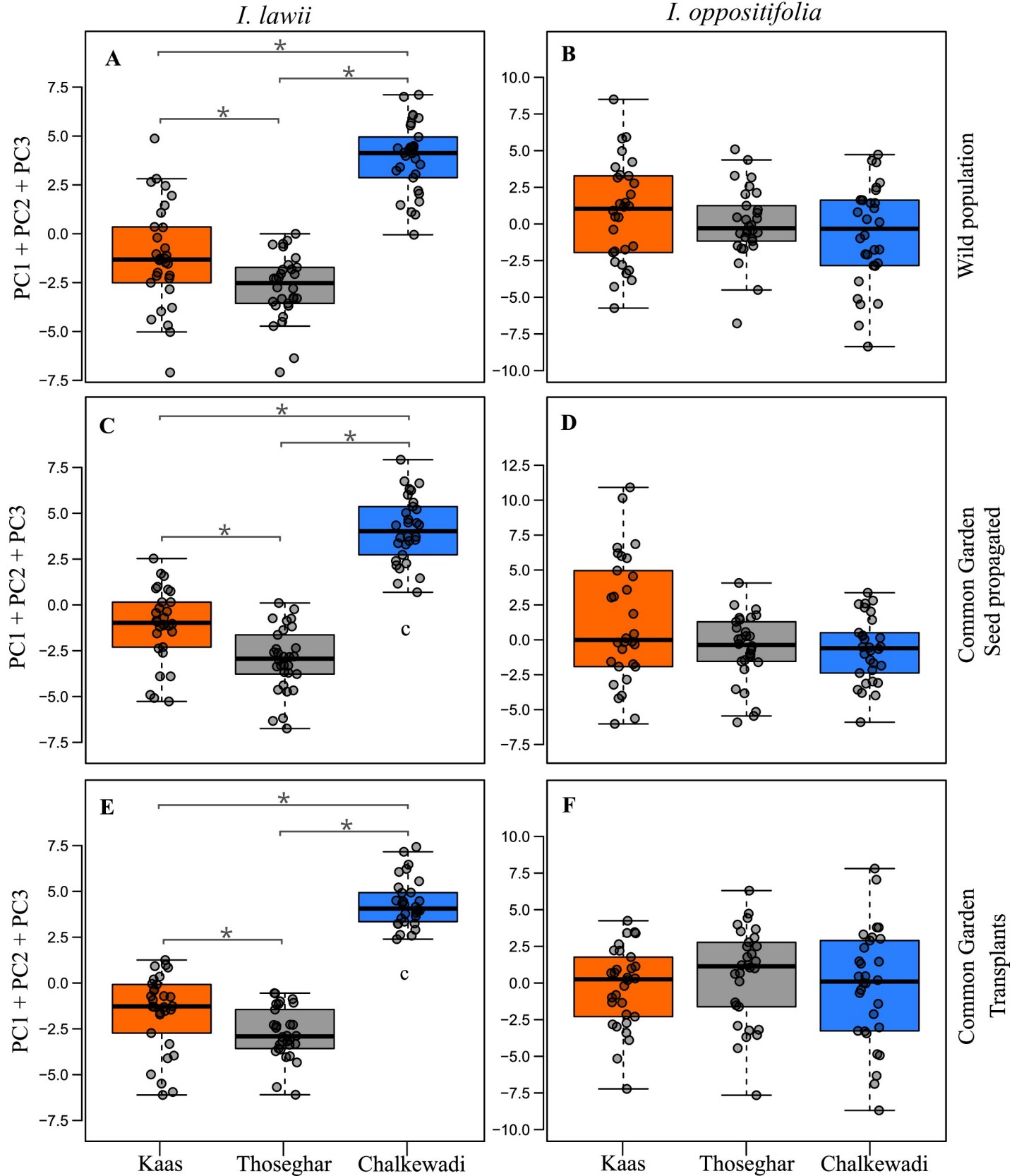

**Fig 4. Floral morphological variation represented by the composite estimates of the first three principal component axes (PC1 + PC2 + PC3) for the floral traits examined in *Impatiens lawii* (A, C, E) and *Impatiens oppositifolia* (B, D, F), for individuals from the three plateaus.** Floral traits of plants from the wild population (A, B) are contrasted with results from plants grown in common garden experiments where plants were propagated from seeds (C, D) or from early stage transplants (E, F). Boxplots with the median and interquartile range for flower width are presented for 30 individuals from each species and plateau. An asterisk indicates a significant difference (p<0.05) between plateaus based on Tukey's post hoc tests.

Fig 5A and 5B). This difference was largely driven by differences in visits by *Apis dorsata*, the Asian Giant Honey Bee, which was one of the most frequent visitors to these two *Impatiens* species. For Lepidoptera, visitation rates were lower for *I. lawii* than for *I. oppositifolia* in both plateaus, but the decrease was greater in Chalkewadi (significant species x plateau interactions, Table 2, Fig 5A and 5B). The results for floral visitation rates at the species/morphotypes level are presented in S5 Table.

## Discussion

*Impatiens lawii* exhibits striking differentiation in floral traits in our study area, the three rocky plateaus of the northern Western Ghats, despite the geographic proximity between the plateaus. The maximum distance between any two plateaus is 17 Km (Kaas-Chalkewadi) while Thoseghar and Chalkewadi are merely 7 Km apart. Yet, floral morphology differs significantly in all pairwise population comparisons (S4 Table; Fig 4A). Flowers from Chalkewadi were

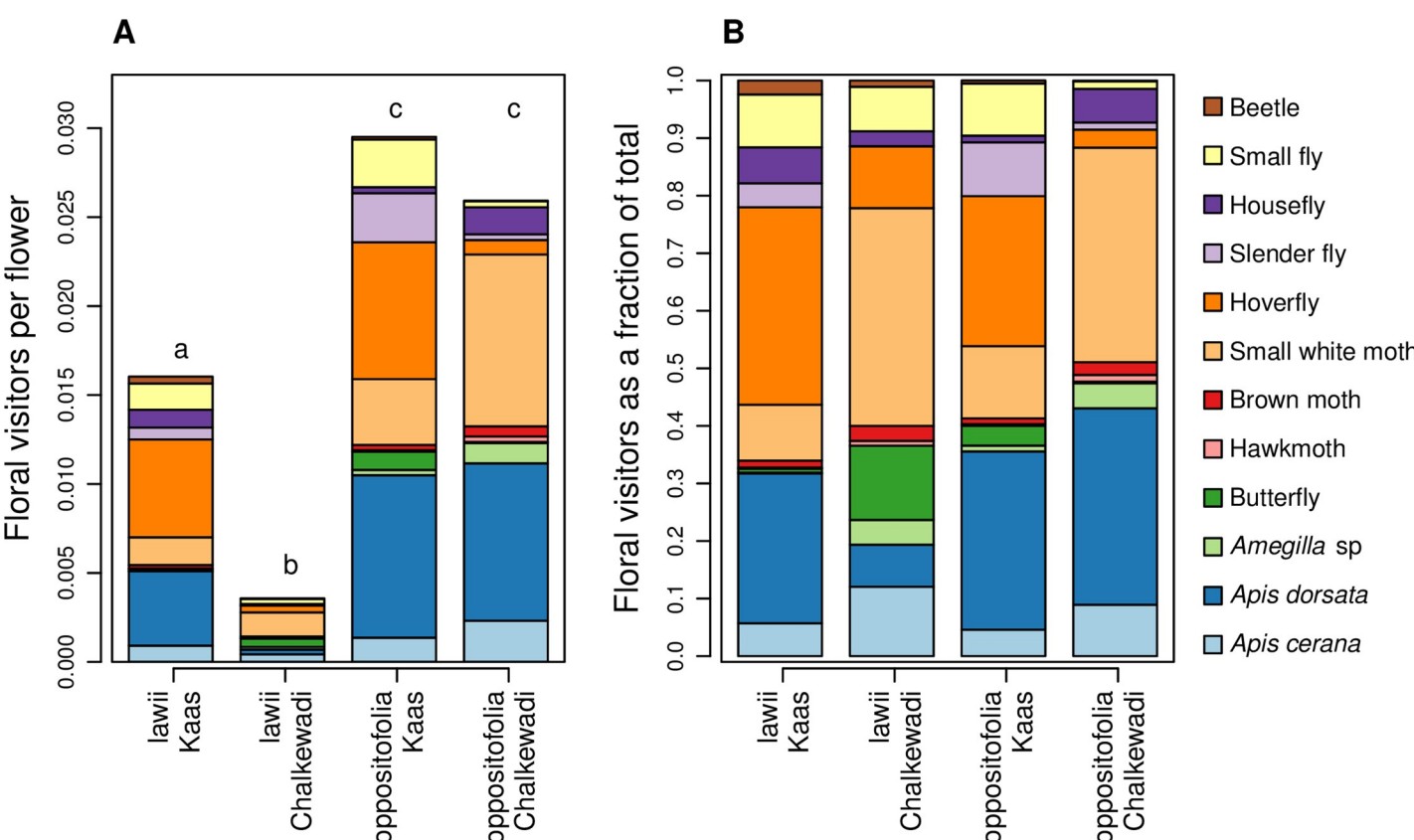

**Fig 5. Floral visitors to *Impatiens lawii* and *I. oppositifolia* in Kaas and Chalkewadi plateaus.** (A) The number of floral visitors belonging to the different putative pollinator groups. The number of floral visitors was normalized by the number of flowers observed. Lower case letters above the bars indicate significant differences (p < 0.05) in total floral visitation rates between plateaus and species based on Tukey's post hoc tests. (B) Floral visitors depicted as the fraction of the total number of visitors.

**Table 2. Variation in floral visitation rates to the *Impatiens* species.** Summary statistics for the effect of plateau and species (fixed variables) and flower number (random variable) are presented.

| | | Estimate | Std. Error | z value | Pr(>\|z\|) | |
|---|---|---|---|---|---|---|
| **a) Total visitors** | | | | | | |
| | Plateau | 1.058 | 0.226 | 4.674 | <0.0001 | *** |
| | Species | 1.602 | 0.210 | 7.641 | <0.0001 | *** |
| | Flower number | 0.001 | 0.000 | 5.105 | <0.0001 | *** |
| | Plateau x Species | -1.397 | 0.284 | -4.912 | <0.0001 | *** |
| **b) Orders** | | | | | | |
| Hymenoptera | Plateau | 1.310 | 0.380 | 3.450 | 0.0006 | *** |
| | Species | 2.050 | 0.341 | 6.014 | <0.0001 | *** |
| | Flower number | 0.001 | 0.000 | 3.395 | 0.0007 | *** |
| | Plateau x Species | -2.028 | 0.436 | -4.650 | <0.0001 | *** |
| Lepidoptera | Plateau | -0.412 | 0.362 | -1.138 | 0.2551 | |
| | Species | 1.513 | 0.314 | 4.820 | <0.0001 | *** |
| | Flower number | 0.000 | 0.000 | 2.320 | 0.0203 | * |
| | Plateau x Species | -1.052 | 0.459 | -2.293 | 0.0218 | * |
| Diptera | Plateau | 1.581 | 0.362 | 4.365 | <0.0001 | *** |
| | Species | 0.519 | 0.361 | 1.438 | 0.1510 | |
| | Flower number | 0.001 | 0.000 | 4.441 | <0.0001 | *** |
| | Plateau x Species | -0.484 | 0.435 | -1.113 | 0.2660 | |
| Coleoptera | Plateau | 1.670 | 0.654 | 2.553 | 0.0107 | * |
| | Species | -0.211 | 0.790 | -0.267 | 0.7891 | |
| | Flower number | 0.002 | 0.001 | 2.911 | 0.0036 | ** |
| | Plateau x Species | -0.625 | 0.957 | -0.653 | 0.5138 | |

Significance is indicated by '***' for p < 0.001, '**' for p < 0.01, and '*' for p < 0.05.

smaller than those from Thosegar and Kaas, while there were no difference between flowers from Kaas and Thoseghar (Fig 3A). The fact that flowers from Kaas and Thosegarh did not differ in size but differed from each other in the multivariate analyses indicates that traits other than size have also diverged between the plateaus. Importantly, floral traits were undifferentiated across plateaus in *I. oppositifolia* (S4 Table; Fig 4B), which co-occurs with *I. lawii* in the plateaus but is a relative generalist with a more continuous distribution between the plateaus. The contrasting patterns between the two species supports our hypothesis that specialization drives stronger floral differentiation, and underscores the influence of habitat specialization on the observed patterns of phenotypic divergences in *I. lawii*.

The patterns of floral trait variation in *I. lawii* were mirrored in both (Transplant and Germination) common garden experiments. Thus, flowers from the transplants and plants reared from seeds originating from Chalkewadi were smaller than those from Kaas and Thosegarh, while there was no size difference between flowers from Kaas and Thoseghar, and all pairwise multivariate comparisons were significantly different (S4 Table; Fig 4C and 4E) These results indicate that the differences in floral morphology in the wild are due to genetic differentiation rather than phenotypic plasticity. In *I. oppositifolia*, there were no differences in floral traits among plateaus in either the Transplant or Germination experiment (S4 Table; Fig 4D and 4F). Therefore, we suggest that the strong degree of specialization of *I. lawii* to rocky plateaus has led to restricted gene flow, and in turn, permitted divergence in floral traits between populations. The persistence of the variation observed in the natural populations in individuals grown in the common garden experiments suggests a genetic basis to the floral traits

examined. Such differences can also arise due to maternal effects which were not accounted for in our study. However, maternal effects are common in early stages in traits such as seed mass and germination, but rare in later stages of plant development in traits such as flower size [58, 59].

One of the primary selective forces on floral morphology is pollination, and, not surprisingly, variation in the pollinator community has been shown to drive diversification of floral traits in many plants [33, 34, 60–65]. *Impatiens* species vary extensively in floral morphology [66–68] and variation in floral traits in this genus has been shown to be related to pollinator visitation [66]. *Impatiens* flowers are zygomorphic, nectar producing and animal pollinated [68]. Additionally, characteristic of this genera, anthers form a fused cap that covers the stigma, physically preventing autogamous self pollination [69]. This fused anther cap is shed before the stigma becomes receptive. Consistent with this dichogamous behaviour and specialized floral morphology, most *Impatiens* species exhibit a high degree of outcrossing [68, 70–72]. Thus, selection by local pollinator communities may drive between-population divergence in floral traits in this group. On the other hand, in species that are primarily outcrossing, greater pollen mediated gene flow may also reduce genetic structure in populations [67].

Our field observations show that floral visitation rates were substantially lower for *I. lawii* in Chalkewadi compared to the same species in Kaas. The lack of any significant differences in visitation rates tor *I. oppositifolia* in the two plateaus implies that abundance of floral visitors was similar for both plateaus, and the reduced visits to *I. lawii* in Chalkewadi was likely a consequence of the differences in floral traits between the Chalkewadi and Kaas *I. lawii* populations. This decrease in the visitation rate was particularly pronounced for *Apis dorsata*, the Asian Giant Honey Bee, which is an important pollinator in rock outcrop communities [73], and is known to be an important and effective pollinator for *Impatiens* species [73–75]. *A. dorsata* was also observed to be one of the most frequent visitors to these species in these plateaus. The differences in floral morphology, particularly the smaller flower size, likely resulted in decreased visits from this large bee to *I. lawii* in Chalkewadi. In addition to being a likely consequence of the observed differences in floral traits, such differences in floral visitation can be important in further reducing gene flow between populations from the different plateaus [76, 77].

Despite the differences between plateaus in the number of floral visitors to *I. lawii*, the species composition of the pollinator communities did not differ strongly. We observed greater abundance of Diptera and Coleoptera in Kaas compared to Chalkewadi, but floral visitors belonging to Hymenoptera and Lepidoptera, which represent important pollinators for *Impatiens* species [66], did not differ between plateaus. This implies that differences in the pollinator communities are unlikely to explain the morphological variation in *I. lawii* between plateaus. However, it is important to note that the species composition of the community of visitors may vary across years and even within the flowering season, and a temporal snapshot of the pollinator community may not reflect selection by the entire pollinator community, particularly over longer periods of time reflective of duration over which these floral traits have diverged. We also acknowledge a limitation of our study is that we have only observed floral visitors and not all visitors may be pollinators. Additionally, even with similar floral visitors, the efficiency of different visitors in successfully pollinating species can differ markedly [78]. We suggest that, given our results, differential selection imposed by contemporary pollinators are unlikely to explain the observed patterns of genetic differentiation, but further studies are needed to confirm differences in the pollinator community and pollination efficiencies.

Although climatic factors are known to influence divergences in floral traits (e.g. [79]), we argue that this is unlikely to be the case for our study populations of *I. lawii*. All three plateaus have similar elevations (Kaas: 1230m; Thoseghar: 1147m; Chalkewadi: 1145m) Further, given

the physical proximity to each other, it is unlikely that climatic conditions differ significantly enough among the plateaus to exert strong differential selection. Among the abiotic factors, edaphic features are most likely to underlie floral trait variation in this species. Both macronutrients [80] and toxic elements [81, 82] are known to influence floral morphology. The soil on rocky plateaus in the northern Western Ghats are rich in metals such as Aluminium and Iron [22], which may directly influence floral morphology. Selection by soil factors can also have pleiotropic effects on floral traits [83].

The isolation-by-distance hypothesis posits that neutral processes, *viz*. genetic drift along with limited gene flow, drives genetic divergences [38]. Early studies [84, 85] on the desert annual plant *Linanthus parryae* suggested that this plant was an excellent empirical example of genetic drift shaping flower colour variation, but subsequent studies [86, 87] have shown that local adaptations better explains floral variation in this species. Indeed, there is little evidence from empirical work that drift has contributed to floral divergences across isolated populations. Studies on *Brassica cretica* [88] and *Iris lutescens* [89] are notable examples supporting drift. We opine that further studies are needed to exclude the role of local selection on flower morphology before invoking genetic drift as an explanation. Reciprocal transplant experiments [90–92] are the gold standard to test hypotheses of local adaptation and will be important to understand floral trait variation in *I. lawii*.

## Correlation of phenotypic divergence with geographic distance

Interestingly, the strongest divergence in floral traits in *I. lawii* was between Thoseghar and Chalkwadi, the two plateaus closest to each other in distance. This was the case both in the wild and in the common garden experiments (Fig 3). The isolation-by-distance model predicts a correlation between the degree of divergence and the distance between plateaus. However, there appears to be no such correlation in the *I. lawii* populations studied here. Studies to estimate the extent of gene flow and population generic structure among *I. lawii* populations promise to shed more light on diversification of this species in the region. In particular, it will be interesting to test whether the extent of genetic differentiation is correlated with phenotypic divergence, or geographic distance.

## Summary and conclusions

We report striking divergences in floral morphology in *I. lawii* in a geographically proximate group of sky island populations. Our experiments show that genetic differentiation, rather than phenotypic plasticity underlies this variation. We are unable to identify selective forces driving local adaptation, and do not exclude genetic drift as an explanation. Reciprocal transplant experiments and analyses of population genetic structure will be particularly illuminating. Our study adds to the growing number of studies demonstrating sky islands in particular, and terrestrial islands in general, as hotspots of diversification.

## Supporting information

**S1 Fig. Variation in floral morphology of *Impatiens lawii* across populations in the Satara plateau cluster.** (A) Map showing the location of the study plateaus within the study area, with the inset indicating the location of the study area within India. Flower images are representative of flower morphology across the three populations. Field images of *I. lawii* from: (B) Kaas; (C) Thoseghar and, (D) Chalkewadi plateaus during the peak flowering season. (PDF)

**S2 Fig. Morphological characters studied in an *Impatiens oppositifolia* flower.** (A) Side view of the whole flower; and, (B) Front view of the flower, depicting the three-dimensional characters measured. (C) Quantification of dissected floral parts. Refer to Table 1 for the details of character abbreviations.
(PDF)

**S3 Fig. Pearson correlations between floral traits in the *I. lawii* and *I. oppositifolia wild populations*.** Graph of a correlation matrix made using corrplot() R function highlighting the most correlated variables in a data table in (A) *I. lawii*; and, (B) *I. oppositifolia*. Positive correlations displayed in shades of blue and negative correlations in red. The colour intensity and the size of the circle are proportional to the value of correlation coefficients which range from -1 to 1. The legend colour on the right side of the correlogram shows the correlation coefficients and the corresponding colours. See Table 1 for variable names.
(PDF)

**S1 Table.** Summary of generalized linear models (GLMs) for flower size (FW) for the wild and common garden experiments for a) *I. lawii*; and, b) *I. oppositifolia*. The GLM coefficient is reported with the estimate, standard error and *p*-value generated from the single-step method. Significance is indicated by *** for p < 0.001 and + for 0.05.
(DOCX)

**S2 Table.** Multiple comparisons of means using Tukey contrasts for flower size (FW) in wild and common garden experiments with: a) *I. lawii*; and, b) *I. oppositifolia*. Tukey post hoc multiple contrasts calculated using multcomp package (Hothorn, Bretz, and Westfall 2008) are reported below with estimate, standard error, z value and *p*-value generated from the single-step method for each pairwise comparison. Significance is indicated by *** for p < 0.001
(DOCX)

**S3 Table. Correlation between floral variables and first three principal component axes.** Eigenvalues, variance percent and cumulative variance percent (CVA) derived from Principal Component Analysis (PCA) of the three populations of *I. lawii* and *I. oppositifolia* in wild and common garden experiments are included. See Table 1 for variable names.
(DOCX)

**S4 Table. Results of pairwise comparison of plateaus based on the scores of principal axis components; PC1, PC1+PC2, and PC1+PC2+PC3.** Tukey post hoc multiple contrasts were calculated using the multcomp package (Hothorn et al. 2008), and are reported below with estimate, standard error, z value and p-value generated from the single-step method for each pairwise comparison. Pr(>|z|) reported are for PC1. Asterisks depict significance with '***' for p < 0.001; '**' for p < 0.01; '*' for p < 0.05; and, 'NS' for p > 0.05.
(DOCX)

**S5 Table. Summary statistics for the effect of plateau and species (fixed variables) and their interaction on rate of floral visits by visitors identified at the level of species/morphotypes in *Impatiens* species.** Flower number was included as a random variable. The regression output table presents coefficients associated with the response variable listed to the left under estimate, standard error associated with these estimates, z value, and the two-tailed p-values that correspond to those z-values and the significance. Asterisks depict significance with '***' for p < 0.001; '**' for p < 0.01; '*' for p < 0.05.
(DOCX)

**S1 Dataset. Larger dataset used to analyse principal component analysis of floral morphology in *Impatiens lawii* and *Impatiens oppositifolia*.** A subset of this data set was also used to plot Fig 2 which represents variation in flower size represented by flower width (FW) across the three wild populations from the three plateaus.
(CSV)

**S2 Dataset. Dataset used to analyse floral morphological variation represented by the composite estimates of the first three principal component axes (PC1 + PC2 + PC3) for the floral traits examined in *Impatiens lawii* and *Impatiens oppositifolia*.** In these analyses, the first three principal components (PC1, PC2 and PC3 were considered as the response variable and plateau as the categorical independent variable.
(CSV)

**S3 Dataset. Dataset used to analyse floral visitors to *Impatiens lawii* and *I. oppositifolia* in Kaas and Chalkewadi.** In these analyses, the response variable included the visitation rates, number of visits contributed by all floral visitors, and the predictor variables included plateau, plant species and flower number.
(CSV)

## Acknowledgments

We thank Gopal Murali and Harshad Vijay Mayekar for their help with the GLM analyses, and Asmi Jezeera, Ashish N. Nerlekar, Ravi Umadi and Sneha Sadanand Joshi for helping with the collection of flower visitation data. We thank two anonymous referees for critical comments that improved the quality of the manuscript.

## Author Contributions

**Conceptualization:** Ullasa Kodandaramaiah, Deepak Barua.

**Data curation:** Aboli Kulkarni.

**Formal analysis:** Sumayya Abdul Rahim, Ullasa Kodandaramaiah, Deepak Barua.

**Funding acquisition:** Ullasa Kodandaramaiah, Deepak Barua.

**Investigation:** Sumayya Abdul Rahim, Ullasa Kodandaramaiah, Deepak Barua.

**Methodology:** Sumayya Abdul Rahim, Ullasa Kodandaramaiah, Aboli Kulkarni, Deepak Barua.

**Project administration:** Sumayya Abdul Rahim, Ullasa Kodandaramaiah, Aboli Kulkarni, Deepak Barua.

**Resources:** Ullasa Kodandaramaiah, Deepak Barua.

**Supervision:** Ullasa Kodandaramaiah, Deepak Barua.

**Validation:** Deepak Barua.

**Visualization:** Sumayya Abdul Rahim, Ullasa Kodandaramaiah, Aboli Kulkarni, Deepak Barua.

**Writing – original draft:** Sumayya Abdul Rahim, Ullasa Kodandaramaiah, Deepak Barua.

**Writing – review & editing:** Sumayya Abdul Rahim, Ullasa Kodandaramaiah, Aboli Kulkarni, Deepak Barua.

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
