## [Decision Letter · Decision Letter 0]

13 May 2021

PONE-D-21-08888

Striking between-population floral divergences in a habitat specialized plant

PLOS ONE

Dear Dr. Ullasa Kodandaramaiah

Thank you for submitting your manuscript to PLOS ONE. After careful consideration, we feel that it has merit but does not fully meet PLOS ONE’s publication criteria as it currently stands. Therefore, we invite you to submit a revised version of the manuscript that addresses the points raised during the review process.

We look forward to receiving your revised manuscript.

Kind regards,

Kleber Del-Claro, PhD

Academic Editor

PLOS ONE

Additional Editor Comments:

Considering the criticism of two expert reviewers we suggest you to provide the changes and consider suggestions to improve the quality of the manuscript for a second round.

Journal Requirements:

3. We note that Figure S1 in your submission contain [map] images which may be copyrighted. All PLOS content is published under the Creative Commons Attribution License (CC BY 4.0), which means that the manuscript, images, and Supporting Information files will be freely available online, and any third party is permitted to access, download, copy, distribute, and use these materials in any way, even commercially, with proper attribution. For these reasons, we cannot publish previously copyrighted maps or satellite images created using proprietary data, such as Google software (Google Maps, Street View, and Earth). For more information, see our copyright guidelines: http://journals.plos.org/plosone/s/licenses-and-copyright.

3.1.    You may seek permission from the original copyright holder of Figure 1 to publish the content specifically under the CC BY 4.0 license. 

3.2.    If you are unable to obtain permission from the original copyright holder to publish these figures under the CC BY 4.0 license or if the copyright holder’s requirements are incompatible with the CC BY 4.0 license, please either i) remove the figure or ii) supply a replacement figure that complies with the CC BY 4.0 license. Please check copyright information on all replacement figures and update the figure caption with source information. If applicable, please specify in the figure caption text when a figure is similar but not identical to the original image and is therefore for illustrative purposes only.

Reviewers' comments:

Reviewer's Responses to Questions

**Comments to the Author**

1. Is the manuscript technically sound, and do the data support the conclusions?

Reviewer #1: Yes

Reviewer #2: Partly

2. Has the statistical analysis been performed appropriately and rigorously? 

Reviewer #1: Yes

Reviewer #2: Yes

3. Have the authors made all data underlying the findings in their manuscript fully available?

Reviewer #1: Yes

Reviewer #2: Yes

4. Is the manuscript presented in an intelligible fashion and written in standard English?

Reviewer #1: Yes

Reviewer #2: Yes

5. Review Comments to the Author

Reviewer #1: The present study presents interesting and robust results about habitat specialization with regard to species diversification. In this respect, the comparative analysis with I. oppositifolia, the experiments in common gardens and the analysis of floral visitors, give support substantiated to the conclusions.

That said, the following analysis basically focuses on the structuring of the manuscript.

Introduction

Some fragments of this item are allocated inappropriately. For example: page 5, lines 86 to 91 ("Our preliminary field observations suggested ..." to "... underlying the patterns of variation), it is characterized more as results than" Introduction "itself.

The third paragraph on page 12, lines 112 to 120 would be more appropriately allocated to the methodological description item.

It is also suggested, at the end of the Introduction item, to clearly and objectively redefine the objectives of this study. For example: "... considering these factors, we aim to quantified pollinator visitation across plateaus through field surveys."

Methods

The sub-item “Study species” is information regarding the introduction. The item “Methods” must be described in a concise, objective and encompassing determinants listed by the authors in a way that understands the tools used for the development of this study.

Considering the methodology used to define the topic “Pollinator visitation”, I recommend changing to flower visitors as presented on Results.

Reviewer #2: General comments

This paper describes the floral traits variation patterns of two species on three plateaus that form sky islands in India. The study highlights the importance of habitat specialization and sky islands in diversification of plants. The paper is innovative in showing that populations of habitat specialized species in sky islands areas, although a very close areas, without any difference on altitude, latitude and climatic factors differ in floral traits and therefore could be led to diversification.

There are three main implicit questions in the manuscript:

“Do floral traits vary similarly for specialized/regional species and non-specialized/widespread distribution species on sky islands?

Which factor best explains the variation on floral traits in these species - phenotypic plasticity or genetic differentiation?

Can floral traits divergence among populations of each species be related to pollinators?”

The aims/hypotheses/questions should be more explicit at the end of introduction section, and the question and hypothesis concerning to floral traits and pollinators need to be defined clearer, as well as the analyses to test it (see comments below).

The authors directly test the phenotypic plasticity and indirectly the genetic differentiation. Although genetic differentiation has not been directly tested (using usual genetic measures such as fixation index Fst or Nei’s Gst), the paper contains a large amount of carefully collected data and suggestive results of genetic differentiation.

Also, there is no mention of the mating system of plant species and its populations. Maybe information about it could be help to explain some results (see details below and the paper: Gamba and Muchhala 2020 10.1111/mec.15575). It should at least be discussed.

As mentioned by the authors in the discussion section, not all floral visitors are pollinators, can be herbivores, thieves, occasional pollinators, effective pollinators, but the term “floral visitors” is being used interchangeably as a synonym for pollinators in the manuscript (see details below). It needs to be revised.

Number of visitors (abundance), number of visits and visitation rate are also merged in the manuscript.

Finally, although populations and species share a similar community of floral visitors (composition), the authors show that the abundance and probably the efficiency change. Despite specific and complex cases herbivores can indirectly influence the floral traits variation, which are mainly affected by abundance and efficiency of pollinators (see Nattero and Cocucci 2007 10.1111/j.1095-8312.2007.00756.x). Furthermore, species seem to show phenotypic integration in floral traits, which is also mediated by pollinators (see Ellis et al 2014 https://doi.org/10.1098/rstb.2013.0563). Thus, pollinator-mediated selection as driving floral traits divergence cannot be considered completely unlikely.

In summary, this paper includes a lot of interesting and new data to the topic. However, clarifing some issues and some additional information are needed to corroborate the conclusions.

Introduction:

The introduction has relevant information that is required to understand the general topic. Some restructuring should be done to clarify the study proposal and to provide smooth reading, and more information added to explain the expected relation between pollinators and floral traits variation in this study.

Line 1 – “for e.g.,” is redundant. Choose between “for example” or “e.g.”. Actually, in this case “i.e.,” is better than others.

Line 65 – The seminal study is missing, which coined the term “sky islands” - Heald, Weldon F. 1951. Sky Islands of Arizona. Natural History, 60: 56-63, 95-96.

Line 88 – Fig S1 – Image from Thoseghar plateau is missing.

Line 122 – Floral traits play important roles in visitation by pollinators, but the floral traits evolution and therefore its differentiation can also be mediated by pollinators.

Regarding structure can be used as a reference: Yun and Kim 2021 https://doi.org/10.1371/journal.pone.0249752 or more specifically, something like that:

Specialized species > sky islands > Western Ghats> Impatiens genus > floral morphology variation > causes of variation > study species and aims/questions/hypotheses (See structure suggested in the attached file)

Methods:

Study species:

– Population sizes? Both species are herbs? Breeding system? Mating system? It is a key information.

Floral variation in natural populations:

Line 145 – “The study species are not cleistogamous”. It can be moved to “study species section”, along with the description of species reproductive system.

Line 146 – We collected fully opened flowers during the peak flowering season It is duplicated (lines 144-145 and 149-151). It can be replaced by:

Whole individuals were collected and stored in sealed plastic bags and one fully opened flower (male or female flowers???) (petals unfolded entirely) was preserved in FAA (Formaldehyde Alcohol Acetic Acid, 10%:50%:5% + 35% water).

Line 156 – Fig 1 and Fig S2 are the same figure.

Line 183 – Table 1 – A column in the table with the association between traits and pollinator efficiency, flower size... would be helpful (it is mentioned only at the results section; lines 294-295).

Transplant experiments:

Line 193 – How many plants were transplanted per species and population?

Pollinator visitation:

Line 207 – Why did you not observe the floral visitors of Thoseghar population?

Line 216 – “All insect species that physically touched the flower were counted as a floral visitor”

As mentioned in the general comments, the shifts in floral traits mainly depend on effectiveness and abundance of pollinators.

In the introduction the authors describe: “In order to understand whether floral trait variation across plateaus in I. lawii has a functional consequence with respect to pollination, we also quantified pollinator visitation across plateaus through field surveys.” Again, the aim needs to be clarified and standardizing the used term (floral visitors or pollinators). If the hypothesis is based on pollinators community, do analysis only with pollinators and change the affirmative to: “All insect species that physically touched the reproductive structures of flower were counted as a pollinator”.

Statistical analyses

Line 224 – What are the vegetative traits?

Results:

Fig 2 – The letters are missing.

Correlation between floral traits and variation in flower size:

Line 277 – The word “letter” is duplicated.

Floral variation in natural populations:

Line 292 – Why did you do a second PCA with all floral traits? Since the first PCA is being used to select the best trait for comparison, FW in this case. (Lines 257-258: “FW was strongly correlated with all other floral characters, and we use FW to represent flower size variation across plateaus.”)

Floral visitation:

Lines 375-377 – This sentence should be moved to discussion section.

Discussion and conclusion:

The authors discussed the results, mentioned the limitations of their study and what could be done in the future to clarify the findings. However, not all statements and conclusions were consistent with introduction and results sections or some relevant information is missing. See suggestions in the comments above.

Line 411 – “floral morphology differs significantly in all pairwise population comparisons (Table S7, Fig 4A-C)”, except for Thoseghar-Kaas.

6. PLOS authors have the option to publish the peer review history of their article (what does this mean?). If published, this will include your full peer review and any attached files.

Reviewer #1: No

Reviewer #2: No

---

## [Author Response · Author response to Decision Letter 0]

25 May 2021

Reviewer1: We thank you for your critical evaluation and comments, which have improved the quality of the manuscript. We have incorporated all your suggestions. 

Reviewer2: We thank you for your critical evaluation and comments, which have improved the quality of the manuscript. We have incorporated all your suggestions.

---

## [Editor Report · Decision Letter 1]

28 May 2021

Striking between-population floral divergences in a habitat specialized plant

PONE-D-21-08888R1

Dear Dr. Ullasa Kodandaramaiah,

We’re pleased to inform you that your manuscript has been judged scientifically suitable for publication and will be formally accepted for publication once it meets all outstanding technical requirements.

Kind regards,

Kleber Del-Claro, PhD

Academic Editor

PLOS ONE

Additional Editor Comments (optional):

This is a simple but significative paper and the authors accepted all the reviewers suggestions and improved the quality of the manuscript significantly. For now, it is accepted for publication.
---

## [Editor Report · Acceptance letter]

18 Jun 2021

PONE-D-21-08888R1 

Striking between-population floral divergences in a habitat specialized plant 

Dear Dr. Kodandaramaiah:

I'm pleased to inform you that your manuscript has been deemed suitable for publication in PLOS ONE. Congratulations! Your manuscript is now with our production department. 

Kind regards, 

on behalf of

Dr. Kleber Del-Claro 

Academic Editor

PLOS ONE